# Why Robust Generalization in Deep Learning is Difficult: Perspective of Expressive Power

**Binghui Li**[1,6,7,*]  **Jikai Jin**[2,*]  **Han Zhong**[3]  **John E. Hopcroft**[4]  **Liwei Wang**[3,5,†]

[1]School of EECS, Peking University
[2]School of Mathematical Sciences, Peking University
[3]Center for Data Science, Peking University   [4]Cornell University
[5]National Key Laboratory of General Artificial Intelligence,
School of Intelligence Science and Technology, Peking University
[6]Peng Cheng Laboratory   [7]Pazhou Laboratory (Huangpu)
`{libinghui,jkjin}@pku.edu.cn, hanzhong@stu.pku.edu.cn,`
`jeh17@cornell.edu, wanglw@cis.pku.edu.cn`

## Abstract

It is well-known that modern neural networks are vulnerable to adversarial examples. To mitigate this problem, a series of robust learning algorithms have been proposed. However, although the robust training error can be near zero via some methods, all existing algorithms lead to a high robust generalization error. In this paper, we provide a theoretical understanding of this puzzling phenomenon from the perspective of expressive power for deep neural networks. Specifically, for binary classification problems with well-separated data, we show that, for ReLU networks, while mild over-parameterization is sufficient for high robust training accuracy, there exists a constant robust generalization gap unless the size of the neural network is exponential in the data dimension $d$. This result holds even if the data is linear separable (which means achieving standard generalization is easy), and more generally for any parameterized function classes as long as their VC dimension is at most polynomial in the number of parameters. Moreover, we establish an improved upper bound of $\exp(\mathcal{O}(k))$ for the network size to achieve low robust generalization error when the data lies on a manifold with intrinsic dimension $k$ ($k \ll d$). Nonetheless, we also have a lower bound that grows exponentially with respect to $k$ — the curse of dimensionality is inevitable. By demonstrating an exponential separation between the network size for achieving low robust training and generalization error, our results reveal that the hardness of robust generalization may stem from the expressive power of practical models.

## 1 Introduction

Deep neural networks have achieved remarkable success in a variety of disciplines including computer vision (Voulodimos et al., 2018), natural language processing (Devlin et al., 2018) as well as scientific and engineering applications (Jumper et al., 2021). However, it is observed that neural networks are often sensitive to small adversarial attacks (Biggio et al., 2013; Szegedy et al., 2013; Goodfellow et al., 2014), which potentially gives rise to reliability and security problems in real-world applications.

In light of this pitfall, it is highly desirable to obtain classifiers that are robust to small but adversarial perturbations. A common approach is to design adversarial training algorithms by using adversarial

---

[*]Equal contribution.
[†]Corresponding author.

36th Conference on Neural Information Processing Systems (NeurIPS 2022).

examples as training data (Madry et al., 2017; Tramèr et al., 2018; Shafahi et al., 2019). Another line of works (Cohen et al., 2019; Zhang et al., 2021a) proposes some provably robust models to tackle this problem. However, while the state-of-the-art adversarial training methods can achieve high robust training accuracy (e.g. nearly $100\%$ on CIFAR-10 (Raghunathan et al., 2019)), all existing methods suffer from large robust test error. Therefore, it is natural to ask what is the cause for such a large generalization gap in the context of robust learning.

Previous works have studied the hardness of achieving adversarial robustness from different perspectives. A well-known phenomenon called the *robustness-accuracy tradeoff* has been empirically observed (Raghunathan et al., 2019) and theoretically proven to occur in different settings (Tsipras et al., 2019; Zhang et al., 2019). Dohmatob (2019) shows that adversarial robustness is impossible to achieve under certain assumptions on the data distribution, while it is shown in Nakkiran (2019) that even when an adversarial robust classifier does exist, it can be exponentially more complex than its non-robust counterpart. Hassani and Javanmard (2022) studies the role of over-parameterization on adversarial robustness by focusing on random features regression models.

At first glance, these works seem to provide convincing evidence that robustness is hard to achieve in general. However, this view is challenged by Yang et al. (2020), who observes that for real data sets, different classes tend to be well-separated (as defined below), while the perturbation radius is often much smaller than the separation distance. As pointed out by Yang et al. (2020), all aforementioned works fail to take this separability property of data into consideration.

**Definition 1.1** (Separated data). *Suppose that $A, B \subset \mathbb{R}^d$ and $\epsilon > 0$. We say that $A, B$ are $\epsilon$-separated under $\ell_p$ norm ($1 \leq p \leq +\infty$) if*

$$\|\boldsymbol{x}_A - \boldsymbol{x}_B\|_p \geq \epsilon, \quad \forall \boldsymbol{x}_A \in A, \boldsymbol{x}_B \in B.$$

Indeed, this assumption is necessary to ensure the existence of a robust classifier. Without this separated condition, it is clear that there is no robust classifier even if a non-robust classifier always exists, as discussed above.

Recently, Bubeck and Sellke (2021) shows that for regression problems, over-parameterization may be necessary for achieving robustness. However, they measure robustness of a model by its training error and Lipschitz constant, which has a subtle difference with *robust test error* (Madry et al., 2017); see the discussions in (Bubeck and Sellke, 2021, Section 1.1).

To sum up the above, although existing robust training algorithms result in low robust test accuracy, previous works do not provide a satisfactory explanation of this phenomenon, since there exists a gap between their settings and practice. In particular, it is not known whether achieving robustness can be easier for data with additional structural properties such as separability (Yang et al., 2020) and low intrinsic dimensionality (Gong et al., 2019).

In this paper, we make an important step towards understanding robust generalization from the viewpoint of neural network expressivity. Focusing on binary classification problems with separated data (cf. Definition 1.1) in $\mathbb{R}^d$, we make the following contributions:

- Given a data set $\mathcal{D}$ of size $N$ that satisfies a separability condition, we show in Section 2 that it is possible for a ReLU network with $\tilde{\mathcal{O}}(Nd)$ weights to robustly classify $\mathcal{D}$. In other words, an over-parameterized ReLU network with reasonable size can achieve $100\%$ robust training accuracy.

- We next consider the robust test error (cf. Definition 3.1). As a warm-up, we show in Section 3 that, in contrast with the robust *training* error, mere separability of data does not imply that low robust test error can be attained by neural networks, unless their size is exponential in $d$. This motivates the subsequent sections where we consider data with additional structures.

- In Section 4, we prove the main result of this paper, which states that for achieving low robust test error, an $\exp(\Omega(d))$ lower bound on the network size is inevitable, even when the underlying distributions of the two classes are linear separable. Moreover, this lower bound holds for arbitrarily small perturbation radius and more general models as long as their VC dimension is at most polynomial in the number of parameters.

- Finally, in Section 5 we consider data that lies on a $k$-dimensional manifold ($k \ll d$), and prove an improved upper bound $\exp(\mathcal{O}(k))$ for the size of neural networks for achieving

Table 1: Summary of our main results.

| Params | Setting | | | |
|---|---|---|---|---|
| | Robust Training | Robust Generalization | | |
| | | General Case | Linear Separable | $k-$dim Manifold |
| Upper Bound | $\mathcal{O}(Nd)$ (Thm 2.2) | $\exp(\mathcal{O}(d))$ (Thm 3.3) | | $\exp(\mathcal{O}(k))$ (Thm 5.5) |
| Lower Bound | $\Omega(\sqrt{Nd})$ (Thm 2.3) | $\exp(\Omega(d))$ (Thm 3.4) | $\exp(\Omega(d))$ (Thm 4.3) | $\exp(\Omega(k))$ (Thm 5.8) |

> low robust test error. Nonetheless, the curse of dimensionality is inescapable – the lower bound is also exponential in $k$.

The upper and lower bounds on network size are summarized in Table 1. Overall, our theoretical analysis suggests that the hardness of achieving robust generalization may stem from the expressive power of practical models.

## 1.1 Implications of our results

Before moving on to technical parts, we would like to first discuss the implications of our results by comparing them to previous works.

Our main result is the exponential lower bound on the neural network size for generalization. First, different from previous hardness results, our result is established for data sets that have desirable structural properties, hence more closely related to practical settings. Note that the separability condition implies that there *exists* a classifier that can perfectly and robustly classify the data i.e. achieve zero robust test error. However, we show that such classifier is hard to approximate using neural networks with moderate size.

Second, it is a popular belief that many real-world data sets are intrinsically low dimensional, although they lie in a high dimensional space. Our results imply that low dimensional structure makes robust generalization possible with a neural network with significantly smaller size (when $k \ll d$). However, the size must still be exponential in $k$.

Finally, we show that there exists an *exponential* separation between the required size of neural networks for achieving low robust training and test error. Based on our results, we conjecture that the widely observed drop of robust test accuracy is not due to limitations of existing algorithms – rather, it is a more fundamental issue originating from the expressive power of neural networks.

## 1.2 Related works

**Robust Generalization.** One surprising behavior of deep learning is that over-parameterized neural networks can generalize well despite their ability to fit random data (Zhang et al., 2017; Belkin et al., 2019). However, in contrast to the standard (non-robust) generalization, for the robust setting, Rice et al. (2020) empirically investigates robust performance of models based on adversarial training methods, which are designed to improve adversarial robustness (Szegedy et al., 2013; Madry et al., 2017), and shows that *robust overfitting* can be observed on multiple datasets. From the theoretical side, Madry et al. (2017) proposes the notion of robust test error to measure the performance of a model under adversarial attacks, and the required sample complexity is studied in various settings (Schmidt et al., 2018; Bhagoji et al., 2019; Dan et al., 2020; Bhattacharjee et al., 2021). In this paper, we mainly focus on this robust generalization gap and provide a theoretical understanding from the perspective of expressive power.

**Robust interpolation.** Bubeck et al. (2021) proposes a conjecture that over-parameterization is necessary for smooth interpolation. Then Bubeck and Sellke (2021) establishes a law of robustness for isoperimetric data. Specifically, they prove an $\tilde{\Omega}(\sqrt{Nd/p})$ Lipschitzness lower bound for smooth interpolation, where $N, d$, and $p$ denote the sample size, the inputs' dimension, and the number of parameters, respectively. This result indicates that over-parameterization may be necessary for robust learning. Zhang et al. (2022) studies many data are needed for robust interpolation. This line

of works focuses on the training error and the worst-case robustness (*i.e.* Lipschitz constant), while we measure robustness via the robust generalization error.

**Memorization power of neural networks.** Our work is related to another line of works (e.g., Baum, 1988; Yun et al., 2019; Bubeck et al., 2020; Zhang et al., 2021a; Rajput et al., 2021; Zhang et al., 2021b; Vardi et al., 2021) on the memorization power of neural networks. Among these works, Yun et al. (2019) shows that a neural network with $\mathcal{O}(N)$ parameters can memorize the data set with zero error, where $N$ is the size of the data set. Under an additional separable assumption, Vardi et al. (2021) derives an improved upper bound of $\tilde{\mathcal{O}}(\sqrt{N})$, which is shown to be optimal. In this work, we show that $\tilde{\mathcal{O}}(Nd)$ parameters is sufficient for achieving low robust training error. This is in contrast with our exponential lower bound for low robust *test* error.

**Function approximation.** Our work is related to a line of works on function approximation via neural networks (e.g., Cybenko, 1989; Hornik, 1991; Lu et al., 2017; Yarotsky, 2017; Hanin, 2019). Yarotsky (2017) is the most related, which shows that the functions in Sobolev spaces can be uniformly approximated by deep ReLU networks. Also related is the studies of using deep ReLU networks to approximate functions supported on low dimensional manifolds (Chui and Mhaskar, 2018; Shaham et al., 2018; Chen et al., 2019). In particular, Chen et al. (2019) proves that any $C^n$ function in Hölder spaces can be $\epsilon$-approximated by the neural network with size $\mathcal{O}(\epsilon^{-k/n})$, where $k$ is the intrinsic dimension of the manifold embedded in $\mathbb{R}^d$. In the robust classification scenario, we can also achieve dimensionality reduction for low-dimensional data.

## 1.3  Notations

Throughout this paper, we use $\|\cdot\|_p$, $p \in [1, +\infty]$ to denote the $\ell_p$ norm in the vector space $\mathbb{R}^d$. For $\boldsymbol{x} \in \mathbb{R}^d$ and $A \subset \mathbb{R}^d$, we can define the distance between $\boldsymbol{x}$ and $A$ as $d_p(\boldsymbol{x}, A) = \inf\{\|\boldsymbol{x} - \boldsymbol{y}\|_p : \boldsymbol{y} \in A\}$. For $r > 0$, $\mathcal{B}_p(\boldsymbol{x}, r) = \left\{\boldsymbol{y} \in \mathbb{R}^d : \|\boldsymbol{x} - \boldsymbol{y}\|_p \leq r\right\}$ is defined as the $\ell_p$-ball with radius $r$ centered at $\boldsymbol{x}$. For a function class $\mathcal{F}$, we use $d_{VC}(\mathcal{F})$ to denote its VC-dimension. A *multilayer neural network* is a function from input $\boldsymbol{x} \in \mathbb{R}^d$ to output $\boldsymbol{y} \in \mathbb{R}^m$, recursively defined as follows:

$$\boldsymbol{h}_1 = \sigma\left(\boldsymbol{W}_1 \boldsymbol{x} + \boldsymbol{b}_1\right), \quad \boldsymbol{W}_1 \in \mathbb{R}^{m_1 \times d}, \boldsymbol{b}_1 \in \mathbb{R}^{m_1},$$
$$\boldsymbol{h}_\ell = \sigma\left(\boldsymbol{W}_\ell \boldsymbol{h}_{\ell-1} + \boldsymbol{b}_\ell\right), \quad \boldsymbol{W}_\ell \in \mathbb{R}^{m_\ell \times m_{\ell-1}}, \boldsymbol{b}_\ell \in \mathbb{R}^{m_\ell}, 2 \leq \ell \leq L - 1,$$
$$\boldsymbol{y} = \boldsymbol{W}_L \boldsymbol{h}_L + \boldsymbol{b}_L, \quad \boldsymbol{W}_L \in \mathbb{R}^{m \times m_L}, \boldsymbol{b}_L \in \mathbb{R}^m,$$

where $\sigma$ is the activation function and $L$ is the depth of the neural network. In this paper, we mainly focus on ReLU networks i.e. $\sigma(x) = \max\{0, x\}$. The size of a neural network is defined as its number of weights/parameters i.e. the number of its non-zero connections between layers.

## 2  Mild Over-parameterized ReLU Nets Achieve Zero Robust Training Error

With access to only finite amount of data, a common practice for learning a robust classifier is to minimize the *robust training error*(defined below). In this section, we show that neural networks with reasonable size can achieve zero robust training error on a finite training set.

**Definition 2.1** (Robust training error). *Given a data set $\mathcal{D} = \{(\boldsymbol{x}_i, y_i)\}_{1 \leq i \leq N}$, $y_i \in \{-1, +1\}$ and an adversarial perturbation radius $\delta \geq 0$, the robust training error of a classifier $f$ is defined as $\hat{\mathcal{L}}_{\mathcal{D}}^{p,\delta}(f) = \frac{1}{N} \sum_{i=1}^{N} \mathbb{I}\{\exists \boldsymbol{x}' \in \mathcal{B}_p(\boldsymbol{x}_i; \delta), \mathrm{sgn}(f(\boldsymbol{x}')) \neq y_i\}$.*

When $\delta = 0$, the definition coincides with the standard training error. In this paper, we mainly focus on the case $p = 2$ and $p = \infty$, but our results can be extended to general $p$ as well.

The following is our main result in this section, which states that for binary classification problems, a neural network with $\widetilde{\mathcal{O}}(Nd)$ weights can perfectly achieve robust classification on a data set of size $N$. The detailed proof is deferred to Appendix B.3.

**Theorem 2.2.** *Suppose that $\mathcal{D} \subset \mathcal{B}_p(0, 1)$ with $p \in \{2, +\infty\}$ consists of $N$ data, and the two classes in $\mathcal{D}$ are $2\epsilon$-separated (cf. Definition 1.1), where $\epsilon \in \left(0, \frac{1}{2}\right)$ is a constant. Let robustness radius $\delta < \frac{1}{2}\epsilon$, then there exists a classifier $f$ represented by a ReLU network with at most*

$$\mathcal{O}\left(Nd \log\left(\delta^{-1}d\right) + N \cdot \mathrm{polylog}(\delta^{-1}N)\right)$$

*parameters, such that* $\hat{\mathcal{L}}_{\mathcal{D}}^{p,\delta}(f) = 0$.

Theorem 2.2 implies that neural networks is quite efficient for robust classification of finite training data. We also derive a lower bound in the same setting, which is stated below. It is an interesting future direction to study whether this lower bound can be achieved.

**Theorem 2.3.** *Let* $p \in \{2, +\infty\}$ *and* $\mathcal{F}_n$ *be the set of functions represented by ReLU networks with at most* $n$ *parameters. For arbitrary* $2\epsilon$-*separated data set* $\mathcal{D}$ *under* $\ell_p$ *norm, if there exists a classifier* $f \in \mathcal{F}_n$ *such that* $\hat{\mathcal{L}}_{\mathcal{D}}^{p,\delta}(f) = 0$, *then it must hold that* $n = \Omega(\sqrt{Nd})$.

The detailed proof of Theorem 2.3 is in Appendix B.4. We leave it as a future direction to study whether this lower bound can be attained. While optimal (non-robust) memorization of $N$ data points only needs constant width (Vardi et al., 2021), our construction in Theorem 2.2 has width $\widetilde{\mathcal{O}}(Nd)$. Therefore, if our upper bound is tight, then Theorem 2.2 can probably explain why increasing the network width can benefit robust training (Madry et al., 2017).

## 3 Hardness of Robust Generalization : A Warm-up

In the previous section, we give an upper bound on the size of ReLU networks to robustly classify finite training data. However, it says nothing about *generalization*, or the robust test error, which is arguably a crucial aspect of evaluating the performance of a trained model. As a warm-up, in this section we first consider the same setting as Section 2 where we only assume the data to be well-separated. We show that in this setting, even achieving high standard test accuracy requires exponentially large neural networks in the worst case, which is quite different from empirical observations. This motivates to consider data with additional structures in subsequent sections.

**Definition 3.1** (Robust test error). *Given a probability measure* $P$ *on* $\mathbb{R}^d \times \{-1, +1\}$ *and a robust radius* $\delta \geq 0$, *the robust test error of a classifier* $f : \mathbb{R}^d \to \mathbb{R}$ *w.r.t* $P$ *and* $\delta$ *under* $\ell_p$ *norm is defined as* $\mathcal{L}_P^{p,\delta}(f) = \mathbb{E}_{(\boldsymbol{x},y) \sim P} \left[ \max_{\|\boldsymbol{x}'-\boldsymbol{x}\|_p \leq \delta} \mathbb{I}\{\text{sgn}(f(\boldsymbol{x}')) \neq y\} \right]$.

In contrast with the training set which only consists of finite data points, when studying generalization, we must consider potentially infinite points in the classes that we need to classify. As a result, we consider two disjoint sets $A, B \in [0,1]^d$, where points in $A$ have label $+1$ and points in $B$ have label $-1$. We are interested in the following questions:

- Does there exists a robust classifier of $A$ and $B$?
- If so, can we derive upper and lower bounds on the size of a neural network to robustly classify $A$ and $B$?

It turns out that, similar to the previous section, the $\epsilon$-separated condition (cf. Definition 1.1) ensures the existence of such a classifier. Moreover, it can be realized by a Lipschitz function. This fact has been observed in Yang et al. (2020), and we provide a different version of their result below for completeness.

**Proposition 3.2.** *For* $2\epsilon$-*separated* $A, B \subset [0,1]^d$ *under* $\ell_p$ *norm with* $p \in \{2, +\infty\}$, *the classifier* $f^*(\boldsymbol{x}) := \frac{d_p(\boldsymbol{x}, B) - d_p(\boldsymbol{x}, A)}{d_p(\boldsymbol{x}, A) + d_p(\boldsymbol{x}, B)}$ *is* $\epsilon^{-1}$-*Lipschitz continuous, and satisfies* $\mathcal{L}_P^{p,\varepsilon}(f^*) = 0$ *for any probability distribution* $P$ *on* $A \cup B$.

Based on this observation, Yang et al. (2020) concludes that adversarial training is not inherently hard. Rather, they argue that current pessimistic results on robust test error is due to the limits of existing algorithms. However, it remains unclear whether the Lipschitz function constructed in Proposition 3.2 can actually be efficiently approximated by neural networks. The following theorem shows that ReLU networks with exponential size is sufficient for as robust classification.

**Theorem 3.3.** *For any two* $2\epsilon$-*separated* $A, B \subset [0,1]^d$ *under* $\ell_p$ *norm with* $p \in \{2, +\infty\}$, *distribution* $P$ *on the supporting set* $S = A \cup B$ *and robust radius* $c \in (0,1)$, *there exists a ReLU network* $f$ *with at most* $\tilde{\mathcal{O}}\left(((1-c)\epsilon)^{-d}\right)$ *parameters, such that* $\mathcal{L}_P^{p,c\epsilon}(f) = 0$.

The detailed proof is deferred to Appendix C.1. Indeed, it is well known that without additional assumptions, an exponentially large number of parameters is also *necessary* for approximating a Lipschitz function (DeVore et al., 1989; Shen et al., 2022). This result motivates us to consider the

second question listed above. The following result implies that even *without* requiring robustness, neural networks need to be exponentially large to correctly classify $A$ and $B$:

**Theorem 3.4.** *Let $\mathcal{F}_n$ be the set of functions represented by ReLU networks with at most $n$ parameters. Suppose that for any $2\epsilon$-separated sets $A, B \subset [0,1]^d$ under $\ell_p$ norm with $p \in \{2, +\infty\}$, there exists $f \in \mathcal{F}_n$ that can classify $A, B$ with zero (standard) test error, then it must hold that $n = \Omega\left((2\epsilon)^{-\frac{d}{2}} \left(d \log (1/2\epsilon)\right)^{-\frac{1}{2}}\right)$.*

Theorem 3.4 implies that mere separability of data sets is insufficient to guarantee that they can be classified by ReLU networks, unless the network size is exponentially large. The detailed proof is in Appendix C.2.

However, one should be careful when interpreting the conclusion of Theorem 3.4, since real-world data sets may possess additional structural properties. Theorem 3.4 does not take these properties into consideration, so it does not rule out the possibility that these additional properties make robust classification possible. Specifically, the joint distribution of data can be decomposed as

$$\mathcal{P}(X, Y) = \underbrace{\mathcal{P}(Y \mid X)}_{\text{labeling mapping}} \underbrace{\mathcal{P}(X)}_{\text{input}},$$

where $\mathcal{P}(X, Y), \mathcal{P}(Y \mid X)$, and $\mathcal{P}(X)$ denote the joint, conditional and marginal distributions, respectively. In subsequent sections, we consider two well-known properties of data sets that correspond to the labeling mapping structure (Section 4) and the input structure (Section 5), respectively, and study whether they can bring improvement to neural networks' efficiency for robust classification.

## 4 Robust Generalization for Linear Separable Data

We have seen that for separated data, if no other structural properties are taken into consideration, even standard generalization requires exponentially large neural networks. However, in practice it is often possible to train neural networks that can achieve fairly high standard test accuracy, indicating a gap between the setting of Section 3 and practice.

This motivates us to consider the following question: assuming that there exists a simple classifier that achieves zero standard test error on the data, is it guaranteed that neural networks with reasonable size can also achieve high *robust* test accuracy?

We give a negative answer to this question. Namely, we show that even in the arguably simplest setting where the given data is linear separable and well-separated (cf. Definition 1.1), ReLU networks still need to be exponentially large to achieve high robust test accuracy.

### 4.1 Main results under the linear separable assumption

Clearly, the robust test error (cf. Definition 3.1) depends on the underlying distribution $P$. We consider a class of data distributions which have bounded density ratio with the uniform distribution:

**Definition 4.1** (Balanced distribution). *Let $S \subset \mathbb{R}^n$ such that there exists a uniform probability measure $m_0$ on $S$. A distribution $P$ on $S$ is called $\mu$-balanced if*

$$\inf \left\{ \frac{P(E)}{m_0(E)} : E \text{ is Lebesgue measurable and } m_0(E) > 0 \right\} \geq \mu.$$

**Remark 4.2.** *Definition 4.1 has also appeared in (Shafahi et al., 2018, Theorem 1), which gives an impossibility result on robust classification, albeit in a completely different setting. Intuitively, it rules out the possibility that data points in certain regions are heavily under-represented.*

The following theorem is the main result of this paper, and the proof sketch is deferred to Section 4.3.

**Theorem 4.3.** *Let $\epsilon \in (0,1)$ be a small constant, $p \in \{2, +\infty\}$ and $\mathcal{F}_n$ be the set of functions represented by ReLU networks with at most $n$ parameters. There exists a sequence $N_d = \Omega\left((2\epsilon)^{-\frac{d-1}{6}}\right), d \geq 1$ and a universal constant $C_1 > 0$ such that the following holds: for any $c \in (0,1)$, there exists two linear separable sets $A, B \subset [0,1]^d$ that are $2\epsilon$-separated under $\ell_p$ norm,*

*such that for any $\mu_0$-balanced distribution $P$ on the supporting set $S = A \cup B$ and robust radius $c\epsilon$ we have*

$$\inf \{ \mathcal{L}_P^{p,c\epsilon}(f) : f \in \mathcal{F}_{N_d} \} \geq C_1 \mu_0.$$

Theorem 4.3 states that the robust test error is lower-bounded by a positive constant $\alpha = C_1 \mu_0$ unless the ReLU network has size larger than $\exp(\Omega(d))$. On the contrary, if we do not require robustness, then the data can be classified by a simple linear function. Moreover, this classifier can be learned with a poly-time efficient algorithm (The detailed proof is in Appendix D.2) :

**Theorem 4.4.** *For any two linear-separable $A, B \subset [0,1]^d$, a distribution $P$ on the supporting set $S = A \cup B$, $\delta > 0$ and $\beta > 0$, let $H$ be the family of $d-$dimensional hyperplane classifiers. Then, there exists a poly-time efficient algorithm $\mathcal{A} : 2^S \to H$, for $N = \Omega(d/\beta^2)$ training instances independently randomly sampled from $P$, with probability $1 - \delta$ over samples, we can use the algorithm $\mathcal{A}$ to learn a classifier $\hat{f} \in F$ such that*

$$\mathcal{L}_P(\hat{f}) \leq \beta,$$

*where $\mathcal{L}_P(f) := \mathbb{P}_{(\boldsymbol{x},y)\sim P}\{y \neq f(\boldsymbol{x})\}$ denotes the standard test error.*

The practical implication of Theorem 4.3 is two-fold. First, by comparing with Theorem 4.4, one can conclude that robust classification may require exponentially more parameters than the non-robust case, which is consistent with the common practice that larger models are used for adversarial robust training. Second, together with our upper bound in Theorem 2.2, Theorem 4.3 implies an *exponential* separation of neural network size for achieving high robust training and test accuracy.

### 4.2 Exponential lower bound for more general models

In general, our lower bounds hold true for a variety of neural network families and other function classes as well, as long as their VC dimension is at most polynomial in the number of parameters, which is formally stated as Theorem 4.5 that can be derived by the proof of Theorem 4.3 directly.

**Theorem 4.5.** *Let $\epsilon \in (0,1)$ be a small constant, $p \in \{2, +\infty\}$ and $\mathcal{G}_n$ be the family of parameterized models with at most $n$ parameters, satisfying the VC-dimension of function family VC-dim$(\mathcal{G}_n)$ is at most $\text{poly}(n)$. Then, there exists a sequence $N_d = \exp(\Omega(d)), d \geq 1$ and a universal constant $C_1' > 0$ such that the following holds: for any $c \in (0,1)$, there exists two linear separable sets $A, B \subset [0,1]^d$ that are $2\epsilon$-separated under $\ell_p$ norm, such that for any $\mu_0$-balanced distribution $P$ on the supporting set $S = A \cup B$ and robust radius $c\epsilon$ we have*

$$\inf \{ \mathcal{L}_P^{p,c\epsilon}(g) : g \in \mathcal{G}_{N_d} \} \geq C_1' \mu_0.$$

In other words, the robust generalization error cannot be lower that a constant $\alpha = C_1' \mu_0$ unless the model, satisfying the property of their VC dimension polynomially bounded by the number of parameters, has exponential larger size. Indeed, this property is satisfied for e.g. feedforward neural networks with sigmoid (Karpinski and Macintyre, 1995) and piecewise polynomial (Bartlett et al., 2019) activation functions. Therefore, our results reveal that the hardness of robust generalization may stem from the expressive power of generally practical models.

### 4.3 Proof sketch of Theorem 4.3

In this subsection, we present a proof sketch for Theorem 4.3 in the $\ell_\infty$-norm case. We only consider $P$ to be the uniform distribution, extending to $\mu_0$-balanced distributions is not difficult, The case of $\ell_2$-norm is similar and can be found in the Appendix.

*Proof Sketch.* Let $K = \lfloor \frac{1}{2\varepsilon} \rfloor$, and $\phi : \{1, 2, \cdots, K\}^{d-1} \to \{-1, +1\}$ be an arbitrary mapping, we define $S_\phi = \left\{ \left( \frac{i_1}{K}, \frac{i_2}{K}, \cdots, \frac{i_{d-1}}{K}, \frac{1}{2} + \epsilon_0 \cdot \phi(i_1, i_2, \cdots, i_{d-1}) \right) : 1 \leq i_1, i_2, \cdots, i_{d-1} \leq K \right\}$, where $\epsilon_0$ is an arbitrarily small constant. The hyperplane $x^{(d)} = \frac{1}{2}$ partitions $S_\phi$ into two subsets, which we denote by $A_\phi$ and $B_\phi$. It is not difficult to check that $A_\phi$ and $B_\phi$ satisfies all the required conditions.

Our goal is to show that there exists some choice of $\phi$ such that robust classification is hard. To begin with, suppose that robust classification with accuracy $1 - \alpha$ can be achieved with at most $M$

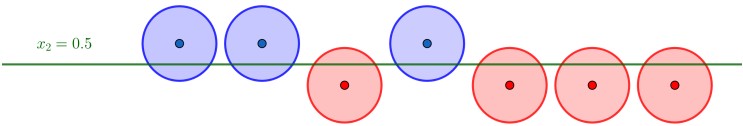

Figure 1: An example of our construction for $d = 2$. We choose $A, B$ as the set of blue points and red points, respectively.

parameters for all $\phi$, then these networks can all be embedded into an *enveloping network* $F_{\boldsymbol{\theta}}$ of size $\mathcal{O}(M^3)$.

Define $\tilde{S} = \left\{ \left( \frac{i_1}{K}, \frac{i_2}{K}, \cdots, \frac{i_{d-1}}{K}, \frac{1}{2} \right) : 1 \leq i_1, i_2, \cdots, i_{d-1} \leq K \right\}$. Robustness implies that for all possible label assignment to $\tilde{S}$, at least $(1 - \alpha)K^{d-1}$ points can be correctly classified by $F_{\boldsymbol{\theta}}$.

If $\alpha = 0$ i.e. perfect classification is required, then we can see that the VC-dim$(F_{\boldsymbol{\theta}}) \geq K^{d-1}$, which implies that its size must be exponential, by applying classical VC-dimension upper bounds of neural networks (Bartlett et al., 2019).

When $\alpha > 0$, we cannot directly use the bound on VC-dimension. Instead, we use a double-counting argument to lower-bound the *growth function* of some subset of $\tilde{S}$.

Let $V = \frac{1}{2}K^{d-1}$. Each choice of $\phi$ corresponds to a $\binom{(1-\alpha)K^{d-1}}{V}$ labelled $V$-subset of $\tilde{S}$ that are correctly classified. There are a total of $2^{K^{d-1}}$ choices of $\phi$, while each labelled $V$-subset can be obtained by at most $2^{K^{d-1}-V}$ different $\phi$. As a result, the total number of labelled $V$-subset correctly classified by $F_{\boldsymbol{\theta}}$ is at least $2^V \binom{(1-\alpha)K^{d-1}}{V}$.

On the other hand, the total number of $V$-subset of $\widetilde{S}$ is $\binom{K^{d-1}}{V}$, thus there must exists a $V$-subset $\mathcal{V}_0 \subset \widetilde{S}$, such that at least

$$\binom{K^{d-1}}{V}^{-1} \cdot 2^V \binom{(1-\alpha)K^{d-1}}{V} \geq \left( \frac{2\left((1-\alpha)K^{d-1} - V\right)}{K^{d-1} - V} \right)^V \geq C_\alpha^{K^{d-1}} \tag{1}$$

different labellings of $\mathcal{V}_0$ are correctly classified by $F_{\boldsymbol{\theta}}$, where $C_\alpha = \sqrt{2(1-2\alpha)} > 1$ for $\alpha = 0.1$. Since (1) provides a lower bound for the growth function, together with the upper bound of growth function in terms of VC-dimension, we can deduce that VC-dim$(F_{\boldsymbol{\theta}}) \geq 0.05K^{d-1}$. Finally, the conclusion follows by applying the VC-dimension bounds in Bartlett et al. (2019). □

**Remark 4.6.** *The connection between VC dimension and approximation error has been explored in a number of previous works (Yarotsky, 2017; Shen et al., 2022) to provide lower bounds on the network size for approximating a given function class. Here we consider the problem of robust classification which is of more practical interest then function approximation, and our main technical contribution is an exponential lower bound on the VC dimension. Our proof formalizes the folklore that adversarial training is hard since it requires a more complicated decision boundary. We note that similar ideas have been used to show benefits of depth in neural networks (Telgarsky, 2016; Liang and Srikant, 2017) but their techniques are restricted to one-dimensional functions.*

## 5  Robust Generalization for Low-Dimensional-Manifold Data

In this section, we focus on refined structure of data's input distribution $\mathcal{P}(X)$. A common belief of real-life data such as images is that the data points lie on a low-dimensional manifold. It promotes a series of methods that are invented to make the dimensionality reduction, including linear dimensionality reduction (e.g., PCA (Pearson, 1901)) and non-linear dimensionality reduction (e.g., $t$-SNE (Hinton and van der Maaten, 2008)). Several works have also empirically verified the belief. Roweis and Saul (2000) and Tenenbaum et al. (2000) have demonstrated that image, speech and other variant form data can be modeled nearly on low-dimensional manifolds. In particular, Wang et al. (2016) studies auto-encoder based dimensionality reduction, and shows that the $28 \times 28 = 784$ dimensional image from MNIST can be reduced to nearly 10 dimensional representations, which corresponds to the intrinsic dimension of the handwritten digital dataset.

Motivated by these findings, in this section, we assume that data lies on a low-dimensional manifold $\mathcal{M}$ embedded in $[0,1]^d$ i.e. $\mathrm{supp}(X) \subset \mathcal{M} \subset [0,1]^d$. We will show a improved upper bound that is exponential in the intrinsic $k$ of the manifold $\mathcal{M}$ instead of the ambient data dimension $d$ for the size of networks achieving zero robust test error, which implies the efficiency of robust classification under the low-dimensional manifold assumption. Also, we point out that the exponential dependence of $k$ is not improvable by establishing a matching lower bound.

## 5.1 Preliminaries

Let $\mathcal{M}$ be a $k-$dimensional compact Riemannian manifold embedded in $\mathbb{R}^d$, where $k$ is the intrinsic dimension ($k \ll d$).

**Definition 5.1** (Chart, atlas and smooth manifold). *A chart for $\mathcal{M}$ is a pair $(U, \phi)$ such that $U \subset M$ is open and $\phi : U \to \mathbb{R}^k$, where $\phi$ is a homeomorphism; An atlas for $M$ is a collection $\{(U_\alpha, \phi_\alpha)\}_{\alpha \in A}$ of pairwise $C^n$ compatible charts such that $\bigcup_{\alpha \in A} U_\alpha = \mathcal{M}$; And we call $\mathcal{M}$ a smooth manifold if and only if $\mathcal{M}$ has a $C^\infty$ atlas.*

**Definition 5.2** (Partition of unity). *A $C^\infty$ partition of unity on a manifold $\mathcal{M}$ is a collection of non-negative $C^\infty$ functions $\rho_\alpha : \mathcal{M} \to \mathbb{R}_+$ for $\alpha \in A$ that satisfy (1) the collection of supports, $\{\mathrm{supp}(\rho_\alpha)\}_{\alpha \in A}$, is locally finite; and (2) $\sum_{\alpha \in A} \rho_\alpha = 1$.*

**Definition 5.3** (Poly-Partitionable). *We call that $\mathcal{M}$ is poly-partitionable if and only if, for a tangent-space-induced atlas $\{(U_\alpha, T_\alpha)\}_{\alpha \in A}$ of $\mathcal{M}$, there exists a particular partition of unity $\{\rho_\alpha\}_{\alpha \in A}$ that satisfies $\rho_\alpha \circ T_\alpha^{-1}$ is a simple piecewise polynomial in $\mathbb{R}^k$, where simple piecewise polynomial is defined as the composite mapping between a polynomial and a size-bounded ReLU network.*

The concept, poly-partitionable, defines a class of manifolds that have simple partition of unity, which is a generalization of some structures in the standard Euclidean space $\mathbb{R}^d$. For example, an explicit construction for low-dimensional manifold $[0,1]^k$ is $\{\phi_m(x)\}$ in Yarotsky (2017), where the coordinate system is identity mapping.

## 5.2 Main results under the low-dimensional manifold assumption

Before giving our main results, we first extend robust classification to the version of manifold.

**Definition 5.4** (Robust classification on a manifold). *Given a probability measure $P$ on $\mathcal{M} \times \{-1, +1\}$ and a robust radius $\delta$, the robust test error of a classifier $f : \mathcal{M} \to \mathbb{R}$ w.r.t $P$ and $\delta$ under $\ell_p$ norm is defined as $\mathcal{L}_{\mathcal{M},P}^{p,\delta}(f) = \mathbb{E}_{(x,y) \sim P} \left[ \max_{x' \in \mathcal{M}, \|x'-x\|_p \leq \delta} \mathbb{I}\{\mathrm{sgn}(f(x')) \neq y\} \right]$.*

Now, we present our main result in this section, which establishes an improved upper bound for size that is mainly exponential in the intrinsic dimension $k$ instead of the ambient data dimension $d$.

**Theorem 5.5.** *Let $\mathcal{M} \subset [0,1]^d$ be a $k-$dimensional compact poly-partitionable Riemannian manifold with the condition number $\tau > 0$. For any two $2\epsilon-$ separated $A, B \subset \mathcal{M}$ under $\ell_\infty$ norm, distribution $P$ on the supporting set $S = A \cup B$ and robust radius $c \in (0,1)$, there exists a ReLU network $f$ with at most*

$$\tilde{\mathcal{O}}\left( \left( (1-c)\,\epsilon/\sqrt{d} \right)^{-\tilde{k}} \right)$$

*parameters, such that $\mathcal{L}_{\mathcal{M},P}^{\infty,c\epsilon}(f) = 0$, where $\tilde{k} = \mathcal{O}(k \log d)$ is almost linear with respect to the intrinsic dimension $k$, only up to a logarithmic factor.*

*Proof sketch.* The proof idea of Theorem 5.5 has two steps. First, we construct a Lipschitz robust classifier $f^*$ in Proposition 3.2. Then, we regard $f^*$ as the target function and use a ReLU network $f$ to approximate it on the manifold $\mathcal{M}$. The following lemma is the key technique that shows we can approximate Lipschitz functions on a manifold by using ReLU networks efficiently.

**Lemma 5.6.** *Let $\mathcal{M} \subset [0,1]^d$ be a $k-$dimensional compact poly-partitionable Riemannian manifold with the condition number $\tau > 0$. For any small $\delta > 0$ and a $L-$lipschitz function $g : \mathcal{M} \to \mathbb{R}$, there exists a function $\tilde{g}$ implemented by ReLU network with at most $\tilde{\mathcal{O}}\left( (\sqrt{d}L/\delta)^{-\tilde{k}} \right)$ parameters, such that $|g - \tilde{g}| < \delta$ for any $x \in \mathcal{M}$, where $\tilde{k}$ is the same as Theorem 5.5.*

By applying the conclusion of Lemma 5.6, we can approximate the $1/\epsilon-$Lipschitz function $f^*$ in Proposition 3.2 via a ReLU network $f$ with at most $\tilde{\mathcal{O}}\left(\exp(\tilde{k})\right)$ parameters, such that the uniform approximation error $\|f - f^*\|_{\ell_\infty(\mathcal{M})}$ at most $1 - c$.

Next, we prove the theorem by contradiction. Assume that there exists some perturbed input $x'$ that is mis-classified and the original input $x$ is in $A$. So we know $f(x') < 0$ and $f^*(x) < \epsilon'$. This impiles $d_\infty(x', A) < d_\infty(x', B) < \frac{1+\epsilon'}{1-\epsilon'} d_\infty(x', A)$. Combined with $d_\infty(x', A) + d_\infty(x', B) \geq d_\infty(A, B) \geq 2\epsilon$, we have $d_\infty(x', A) > (1 - \epsilon')\epsilon = c\epsilon$, which is a contradiction. $\qquad\square$

**Remark 5.7.** *Chen et al. (2019) studies network-based approximation on smooth manifolds, and also establishes an $\mathcal{O}(\delta^{-k})$ bound for the network's size. However, different from their setting where the approximation error $\delta$ goes to zero, it is reasonable that the separated distance $\epsilon$ and robust radius $c$ are constants in our setting. If we simply follow their proofs, we can only obtain the bound $\mathcal{O}((\delta/\mathcal{C}_\mathcal{M})^{-k})$ where $\mathcal{C}_\mathcal{M}$ also grows exponentially with respect to $k$, which further implies that the final result will be roughly $\exp(\mathcal{O}(k^2))$. This bound is too loose, especially when $k \approx \sqrt{d}$. To this end, we propose a novel approximation framework so as to improve the bound to $\exp(\mathcal{O}(k))$, which is presented as Lemma 5.6. And the detailed proof of Lemma 5.6 is deferred to Appendix E.1.*

Although we have shown that robust classification will be more efficient when data lies on a low-dimensional manifold, there is also a curse of dimensionality, i.e., the upper bound for the network's size is almost exponential in the intrinsic dimension $k$. The following result shows that the curse of dimension is also inevitable under the low-dimensional manifold assumption.

**Theorem 5.8.** *Let $\epsilon \in (0, 1)$ be a small constant. There exists a sequence $\{N_k\}_{k \geq 1}$ that satisfies $N_k = \Omega\left((2\epsilon\sqrt{d/k})^{-\frac{k}{2}}\right)$. and a universal constant $C_1 > 0$ such that the following holds: let $\mathcal{M} \subset [0,1]^d$ be a complete and compact $k-$dimensional Riemannian manifold with non-negative Ricci curvature , then there exists two $2\epsilon$-separated sets $A, B \subset \mathcal{M}$ under $\ell_\infty$ norm, such that for any $\mu_0-$balanced distribution $P$ on the supporting set $S = A \cup B$ and robust radius $c \in (0, 1)$, we have $\inf\{\mathcal{L}_P^{\infty,c\epsilon}(f) : f \in F_{N_k}\} \geq C_1\mu_0$.*

In other words, the robust test error is lower-bounded by a positive constant $\alpha = C_1\mu_0$ unless the neural network has size larger than $\exp(\Omega(k))$. The detailed proof of Theorem 5.8 is presented in Appendix E.4.

# 6 Conclusion

This paper provides a new theoretical understanding of the gap between the robust training and generalization error. We show that the ReLU networks with reasonable size can robustly classify the finite training data. On the contrary, even with the linear separable and well-separated assumptions, ReLU networks must be exponentially large to achieve low robust generalization error. Finally, we consider the scenario where the data lies on the low dimensional manifold and prove that the ReLU network, with a size exponentially in the intrinsic dimension instead of the inputs' dimension, is sufficient for obtaining low robust generalization error. We believe our work opens up many interesting directions for future work, such as the tight bounds for the robust classification problem, or the reasonable assumptions that permit the polynomial-size ReLU networks to achieve low robust generalization error.

## Acknowledgement

We thank all the anonymous reviewers for their constructive comments. This work is supported by National Science Foundation of China (NSFC62276005), The Major Key Project of PCL (PCL2021A12), Exploratory Research Project of Zhejiang Lab (No. 2022RC0AN02), and Project 2020BD006 supported by PKUBaidu Fund. Binghui Li is partially supported by National Innovation Training Program of China. Jikai Jin is partially supported by the elite undergraduate training program of School of Mathematical Sciences in Peking University.

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
