# OpenReview forum: "Why Robust Generalization in Deep Learning is Difficult: Perspective of Expressive Power"
_NeurIPS.cc/2022/Conference — NeurIPS 2022 Accept_

### Official Review · Reviewer_pcp1 · 2022-07-06

**Rating:** 7
**Confidence:** 3
**Soundness:** 4 excellent
**Presentation:** 3 good
**Contribution:** 3 good

**Summary:**

The paper analyses the number of parameters of a ReLU neural network required for robust generalization. The results are derived by defining a function that would robustly classify (train and test) and to show that this function can be approximated by a ReLU network with a certain amount of parameters. The paper provides interesting insights into the required level of overparameterization both for achieving low robust training and test error, in particular showing that robust test error in general requires a number of parameters exponential in the data dimension. If data lies on a low-dimensional manifold, this can be mitigated, though.

The paper is well-written, the theoretical results are sound, interesting, and relevant. I vote for acceptance.

**Questions:**

- After Def. 1.1. the authors remark that data needs to be $\epsilon$-seperated to ensure the existence of a robust classifier. Although it might seem trivial, for me it would have been helpful to formalize this result. I honestly was thinking for a bit how a suitable representation of the data would make it $\epsilon$-separated until I realized that no representation can actually solve the problem, since we assume perturbations on the input data so that the required radius is transformed by the representation as well.
- Generalization bounds based on flatness of the loss surface (cf. Thm. 6 in [1]) guarantee low test error (the bound in [1] is also exponential in the dimension), but flatness of the loss surface also seems to imply a robust test error as defined in Def. 3.1, i.e., robustness in expectation. The bound in [1] is exponential in the dimension of the representation, not the input data, though. Is this related to the results on learning from a low-dimensional manifold?
- Given the papers results on low-dimensional maniforlds, it would be great to also study the required number of parameters to project on such a manifold. To me it seems possible that a proper representaion reduces the overall number of parameters required. More formally, assume a robust classifier $f=\psi\circ\phi$ with representation $\phi$ and classifier $\psi$, if $\phi$ projects on a low-dimensional manifold and the projection requires only $n_1$ parameters and the manifold has an intrinsic dimension of $\mathcal{O}(\log n_2 / \log d)$ (following the result in Thm 5.5), then the network would have $n_1+n_2$ parameters which could be less than the exponential number of parameters predicted by Thm. 4.3.

[1] Petzka, Henning, et al. "Relative flatness and generalization." Advances in Neural Information Processing Systems 34 (2021): 18420-18432.


**Limitations:**

The theoretical results are not empirically verified.

**Strengths And Weaknesses:**

Strength:
- strong theoretical results
- relevant analysis

Weakness:
- no empirical validation of the theoretical results

---

> ### Author Response · Authors · 2022-07-31
> **Response to Reviewer pcp1**
>
> We would like to thank the reviewer for the thoughtful review and positive feedback. Below are our responses to the questions and suggestions raised by the reviewer.
>
> $\textbf{Regarding empirical validation.}$ In this paper, we mainly focus on the robust generalization gap theoretically, and provide a theoretical understanding of this phenomenon from the perspective of expressive ability. More empirical evidence about the robust generalization gap (robust overfitting) has been systematically shown in the previous paper[1]. We will consider adding a simulation in the revision. Thanks for the reviewer's suggestion.
>
> $\textbf{Regarding the necessity of ε-separation.}$ As the review suggests, it will be clearer that data needs to be ϵ-separated to ensure the existence of a robust classifier if we formalize the issue. We will add some more detailed remarks about it in the revision. Thanks for the reviewer's suggestion.
>
> $\textbf{Regarding the generalization bound based on flatness.}$ The paper[2] proves a generalization bound based on the relative flatness of the loss landscape, and the bound in [2] is exponential in the dimension of the representation, which implies that a good low-dimensional representation of data will improve the generalization. Indeed, low-dimensional manifold data has a natural low-dimensional representation via approximating chart mapping $\phi : U \rightarrow \mathbb{R}^{k}$ (Def 5.1 in Section 5) by neural networks, where $k$ is the data intrinsic dimension. Therefore, our results about low-dimensional manifold settings are potentially related to the results about relative flatness and generalization in the paper[2]. Thanks for pointing out the paper, and we will discuss about it in the related works.
>
> $\textbf{Regarding projecting on a low-dimensional manifold.}$ As the reviewer insightfully suggests, it is very interesting to study the required number of parameters to project on a low-dimensional manifold, which may help us to improve the bound of the network size for achieving low robust generalization error. We will study this in the future, and thanks for the reviewer's suggestion again.
>
> [1] Rice, L., Wong, E., & Kolter, Z. (2020, November). Overfitting in adversarially robust deep learning. In International Conference on Machine Learning (pp. 8093-8104). PMLR.
>
> [2] Petzka, Henning, et al. "Relative flatness and generalization." Advances in Neural Information Processing Systems 34 (2021): 18420-18432.

---

> > ### Comment · Reviewer_pcp1 · 2022-08-05
> > **Response to authors**
> >
> > Dear authors,
> >
> > Thank you for your reply. Just to clarify: I do not think that your paper needs an empirical validation. I suggest to simply point towards the empirical evidence in [1] that you mention.
> >
> >
> > Cheers

---

> > > ### Author Response · Authors · 2022-08-05
> > > **Thank you**
> > >
> > > Thanks for the clarification. We will incorporate your suggestion in the revision.

---

### Official Review · Reviewer_eEJn · 2022-07-12

**Rating:** 6
**Confidence:** 3
**Soundness:** 3 good
**Presentation:** 3 good
**Contribution:** 3 good

**Summary:**

This paper studies the approximation power of feed-forward neural networks with ReLu activations (FFNs) to approximate robust functions. Robustness of a model is neccessary to avoid adverserial attacks. However, training a robust model seems to be hard. Different from prior studies on algorithmic aspects, this paper investigates approximation ability of FFNs. Four main results are presented, and indicate both possitive and negative aspects. The possitive result is that there exists a FFN with moderate size that has zero training error. The negative results are that, to ensure small test error, one may need FNNs with exponential size in dimensionality of data. This result also holds even under a separability condition. The authors provide both upper and lower bounds for the size of those FNNs. Those bounds can be improved when the data lies on a low dimensional manifold.

**Questions:**

The depth often plays an important role in approximation power of neural networks in non-robust settings. How about the role of depth in robust training?

**Strengths And Weaknesses:**

Strengths:
- This paper provides a novel view about the hardness of training robust models, by investigating approximation ability.
- The lower bounds on the network size seem to be novel and provide evidences for why training robust models are hard.


Weaknesses:
- The scope of this paper is unclear. This paper only studies FFNs with ReLu activations, and their results hold true for those families. However the current writing may make readers to confuse with general neural networks. Some other network families may have stronger ability, and the results of this paper may not hold true. The authors should indicate this point clearly from the beggining.
- The proof of the main theorems (4.3) seems to borrow some ideas or results from (Shen et al. 2022). One example is the connection between VC dimension and approximation error. However this point is not indicated in the paper.
- The depth of FNNs seems not to be important in their bounds.

Minor: family F_{N_d} in Theorem 4.3 is not defined explicitly.

---

> ### Author Response · Authors · 2022-07-31
> **Response to Reviewer eEJn**
>
> We would like to thank the reviewer for the thoughtful review and positive feedback. Below are our responses to the questions and suggestions raised by the reviewer.
>
> $\textbf{Regarding general neural networks.}$ We have mentioned in the abstract (e.g. Line 7) and main theorems (e.g. Theorems 3.4 and 4.3) that we focused on ReLU networks, and we will clarify this more in the revision. We expect that our lower bounds hold true for a variety of neural network families and other function classes as well, as long as their VC dimension is at most polynomial in the number of parameters.  This property is satisfied for e.g. feedforward neural networks with sigmoid[1] and piecewise polynomial[2] activation functions. This is because we prove that any function class that can achieve robust generalization must have a VC dimension at least exponential in $d$.  We think it is an interesting future direction to explore the robust generalization ability of more general neural networks, and we'll follow the reviewer's advice to add some discussions in the introduction part.
>
> $\textbf{Regarding the proof idea of Theorem 4.3.}$ As noticed by the reviewer, the connection between VC dimension and approximation error is not new. Our main technical contribution here is that a neural network that can achieve robust generalization must have an exponentially large VC dimension. This result allows us to derive an exponential lower bound on the size of ReLU networks. We'll add some discussions on related works in an updated version of this paper.
>
> $\textbf{Regarding the role of depth.}$ For our upper bound on the network size for robust training, we construct a ReLU network whose depth scales logarithmically in dataset size and dimension. This is not surprising, since it is shown in different settings [3,4] that a constant-depth neural network with moderate size can perfectly memorize a finite data set.  Nonetheless, we leave it as a future question whether increasing depth can lead to a tighter upper bound.
>
> [1] Karpinski M, Macintyre A. Polynomial bounds for VC dimension of sigmoidal neural networks./Proceedings of the twenty-seventh annual ACM symposium on Theory of computing. 1995: 200-208.
>
> [2] Bartlett P L, Harvey N, Liaw C, et al. Nearly-tight VC-dimension and pseudodimension bounds for piecewise linear neural networks. The Journal of Machine Learning Research, 2019, 20(1): 2285-2301.
>
> [3] Bubeck S, Eldan R, Lee Y T, et al. Network size and size of the weights in memorization with two-layers neural networks. Advances in Neural Information Processing Systems, 2020, 33.
>
> [4] Yun C, Sra S, Jadbabaie A. Small ReLU networks are powerful memorizers: a tight analysis of memorization capacity. Advances in Neural Information Processing Systems, 2019, 32.

---

### Official Review · Reviewer_diHV · 2022-07-20

**Rating:** 7
**Confidence:** 4
**Soundness:** 3 good
**Presentation:** 4 excellent
**Contribution:** 3 good

**Summary:**

This paper presents lower and upper bounds for the number of parameters needed in a ReLU neural network to learn a robust classifier on well-separated data. The analysis considers robust training and testing error respectively, as well as general and linear separability and intrinsic low-dimensional structural assumption in terms of the structure of the dataset.

**Questions:**

In Def. 3.1, I assume you meant $\mathrm{sgn}(f(x')) \neq y$ instead of $f(x') \neq y$?


**Limitations:**

This paper is theoretical in nature, hence it should not have any potential adverse social impact.

**Strengths And Weaknesses:**

This paper addresses the underlying reason of the decay of robust generalization error observed in real-world experiments through analyzing the expressive capability of ReLU neural networks and thus establishes the desired hardness result based on the shown exponential lower-bound on network size, instead of usual algorithmic approaches which mainly constructs sample complexity or SQ super-poly lower-bounds for specific problem instances. Hence in terms of originality this paper potentially opens up interesting new techniques for investigating similar problems. The paper is also written clearly and its argument meets the desired technical quality for a deep learning theory paper.
However, it is unclear whether the bounds the paper presents are tight, which will make the results even more compelling if their optimality can be established as well. Another interesting analysis that is good to have is identifying in addition to data being well-separated, under what extra conditions is achieving low robust test error attainable with non-exponential network size (or even slightly super-polynomial).

---

> ### Author Response · Authors · 2022-07-31
> **Response to Reviewer diHV**
>
> We would like to thank the reviewer for the thoughtful review and positive feedback. Below are our responses to the questions and suggestions raised by the reviewer.
>
> $\textbf{Regarding the statement of Definition 3.1.}$ Thank the reviewer for noticing this typo. As the reviewer points out, in Def. 3.1, the input of the indicator $\mathbb{I}${ ... } should be $\operatorname{sgn}(f(x′))\ne y$ rather than $f(x′)\ne y$ because we consider binary classification tasks in this paper.
>
> $\textbf{Regarding the optimality of the bounds.}$ As the reviewer insightfully suggests, it is interesting to explore the optimality of the bounds we derived in this paper. For the robust training, combined with the lower bound of VC dimension mentioned in the proof of Thm 2.3, we know that the upper bound in Thm 2.2 is nearly tight, only up to a logarithmic factor, when we restrain the depth of networks is not more than the logarithmic order. We leave it as a future direction whether the upper bound can be improved if we increase the depth.
>
> $\textbf{Regarding extra conditions.}$ We are also seeking more extra reasonable assumptions for data structures in order to permit the practical models with a non-exponential size to achieve low robust generalization error. Thanks for the reviewer's suggestion again.

---

### Meta-Review · Area_Chair_SkMN · 2022-08-27

**Recommendation:** Accept
**Confidence:** Certain

**Metareview:**

This paper gives new lower bounds and upper bounds on the size of a feedforward ReLU network needed for robust generalization (and not just robust training error. In particular, they give an exponential lower bound on the size of the network, even for separable data. The reviewers were in agreement about the strengths the paper. This points one of the challenges in obtaining neural networks with robust test error.

**Award:**

No

---

### Decision · Program_Chairs · 2022-09-14

Accept